# An XRCC4 mutant mouse, a model for human X4 syndrome, reveals interplays with Xlf, PAXX, and ATM in lymphoid development

Benoit Roch[1,2†], Vincent Abramowski[1,2†], Olivier Etienne[3], Stefania Musilli[1,2], Pierre David[4], Jean-Baptiste Charbonnier[5], Isabelle Callebaut[6], François D Boussin[3], Jean-Pierre de Villartay[1,2]*

[1]Université de Paris, Imagine Institute, Laboratory "Genome Dynamics in the Immune System", INSERM UMR 1163, F-75015, Paris, France; [2]Equipe Labellisée Ligue Nationale Contre le Cancer, F75015, Paris, France; [3]Université de Paris and Université Paris-Saclay, Inserm, LRP/iRCM/IBFJ CEA, UMR Stabilité Génétique Cellules Souches et Radiations, F-92265, Fontenay-aux-Roses, France; [4]Université de Paris, Imagine Institute, Transgenesis facility, INSERM UMR 1163, F-75015, Paris, France; [5]Institute for Integrative Biology of the Cell (I2BC), Institute Joliot, CEA, CNRS, Univ. Paris-Sud, Université Paris-Saclay, F-91198, Gif-sur-Yvette Cedex, France; [6]Sorbonne Université, Muséum National d'Histoire Naturelle, CNRS UMR 7590, Institut de Minéralogie, de Physique des Matériaux et de Cosmochimie, F-75005, Paris, France

*For correspondence:
jean-pierre.de-villartay@inserm.fr

†These authors contributed equally to this work

Competing interest: The authors declare that no competing interests exist.

**Abstract** We developed an $Xrcc4^{M61R}$ separation of function mouse line to overcome the embryonic lethality of Xrcc4-deficient mice. $XRCC4^{M61R}$ protein does not interact with Xlf, thus obliterating XRCC4-Xlf filament formation while preserving the ability to stabilize DNA ligase IV. $X4^{M61R}$ mice, which are DNA repair deficient, phenocopy the $Nhej1-/-$ (known as $Xlf -/-$) setting with a minor impact on the development of the adaptive immune system. The core non-homologous end-joining (NHEJ) DNA repair factor XRCC4 is therefore not mandatory for V(D)J recombination aside from its role in stabilizing DNA ligase IV. In contrast, $Xrcc4^{M61R}$ mice crossed on $Paxx-/-$, $Nhej1-/-$, or $Atm-/-$ backgrounds are severely immunocompromised, owing to aborted V(D)J recombination as in $Xlf-Paxx$ and $Xlf-Atm$ double Knock Out (DKO) settings. Furthermore, massive apoptosis of post-mitotic neurons causes embryonic lethality of $Xrcc4^{M61R}$ $-Nhej1-/-$ double mutants. These in vivo results reveal new functional interplays between XRCC4 and PAXX, ATM and Xlf in mouse development and provide new insights into the understanding of the clinical manifestations of human $XRCC4$-deficient condition, in particular its absence of immune deficiency.

## Introduction

Living organisms face DNA double-strand breaks (DSBs), the most toxic DNA lesions, from random or programmed (prDSBs) origins (*Betermier et al., 2020*), such as during the development of the adaptive immune system through V(D)J recombination. V(D)J recombination results in the somatic rearrangement of variable (V), diversity (D), and joining (J) elements of antigen receptor loci in T- and B-cell precursors (*Jung et al., 2006*). It is initiated by the domesticated transposase Recombination-Activating Genes 1 and 2 factors (RAG1/2), which introduce prDSBs at the border of V, D, and J elements within recombination signal sequences (RSS). The non-homologous end-joining (NHEJ) machinery is the sole DNA repair pathway to cope with these lymphoid-specific prDSBs. Briefly, the

NHEJ pathway is signaled by the Ataxia-Telangiectasia mutated (ATM) protein (*Arya and Bassing, 2017*; *Bassing et al., 2002*) and prompted by the Ku70-Ku80 complex that recognizes broken DNA ends and recruits the DNA-dependent protein kinase-catalytic subunit (DNA-PKcs), which further activates the pathway (*Sibanda et al., 2017*). Broken DNA ends are processed by Artemis endo/ exonuclease, which specifically opens the RAG1/2-generated hairpins at DNA ends during V(D) J recombination (*Chang et al., 2017*). DNA ligase IV (Lig4), X-ray repair cross-complementation 4 (XRCC4, or X4), XRCC4-like factor (Xlf, also known as Cernunnos), and Paralog of XRCC4 and Xlf (PAXX) (*Craxton et al., 2015*; *Ochi et al., 2015*; *Xing et al., 2015*) ensure ligation of broken ends (*Lieber, 2010*). Lig4 and XRCC4 are critical factors during development apart from their essential function during V(D)J recombination, and both *Lig4-/-* and *Xrcc4-/-* mice suffer late embryonic lethality caused by apoptosis of post-mitotic neurons (*Barnes et al., 1998*; *Frank et al., 1998*; *Shull et al., 2009*). Likewise, human patients with LIG4 syndrome present with dwarfism, microcephaly, and various degrees of immunodeficiency caused by hypomorphic mutations in the *Lig4* gene (*O'Driscoll et al., 2001*; *Staines Boone et al., 2018*).

In most of the cases, defects in core NHEJ factors result in severe combined immunodeficiency (SCID), owing to aborted V(D)J recombination both in humans and animal models (*de Villartay et al., 2003*). NHEJ deficiency also results in genetic instability with the development of Pro-B cell lymphomas harboring chromosomal translocations when associated with *Trp53-/-* (*Alt et al., 2013*). *Xlf/Cernunnos*-deficient patients also present with developmental features including dwarfism, micro- cephaly, and combined immunodeficiency (CID) (*Buck et al., 2006*). However, V(D)J recombination is not severely affected nor in human (*Buck et al., 2006*; *Recio et al., 2018*; *van der Burg and Gennery, 2011*) or murine (*Li et al., 2008*; *Roch et al., 2019*; *Vera et al., 2013*) settings of *Xlf* deficiency. This paradoxical situation is a consequence of the functional redundancy between Xlf and several DNA repair factors, including PAXX, ATM, H2A.X, MDC1, MRI, and 53BP1 as revealed by the complete V(D) J recombination defect in combined deficient settings (*Abramowski et al., 2018*; *Beck et al., 2019*; *Hung et al., 2018*; *Musilli et al., 2020*; *Oksenych et al., 2012*; *Zha et al., 2011*) as well as RAG2 itself (*Lescale et al., 2016a*). Consistent with their overall efficient V(D)J recombination, *Nhej1-Trp53* DKO mice indeed do not develop Pro-B cell lymphomas (*Vera et al., 2013*).

To account for this functional redundancy, we proposed a model in which prDSBs may benefit from evolutionary conserved DNA repair mechanisms as to avoid their intrinsic oncogenic potential (*Beter- mier et al., 2020*). In the particular context of V(D)J recombination, this mechanism would operate through a redundant 'double DNA repair synapse,' which strictly operates during V(D)J recombination to prevent genomic instability, but not in case of genotoxic-induced DNA damage (*Abramowski et al., 2018*; *Lescale et al., 2016a*). The 'two-synapses' model accounts for the absence of V(D)J recombination defect in the absence of Xlf. One essential actor of the 'two-synapses' model is the RAG2 factor itself, which together with RAG1 is known to remain on DNA broken ends during V(D) J recombination, forming the so-called post-cleavage complex (PCC) (*Schatz and Swanson, 2011*). We previously established that the C-terminus of RAG2 is determinant in complementing the lack of Xlf (*Lescale et al., 2016a*). Indeed, the combined absence of the C-terminus of RAG2 and Xlf results in SCID mice owing to a complete block of V(D)J recombination. In the 'two-synapses' model, the first synapse would be mediated by RAG2, PAXX, and ATM signaling as suggested by efficient V(D) J recombination in multiply deficient v-Abl transformed Pro-B cells (*Lescale et al., 2016a*; *Lescale et al., 2016b*);. The second synapse is constituted by the Xlf-X4 filament or the *bona fide* NHEJ core complexes, the structure of which was recently resolved through cryoelectron microscopy (cryo-EM) (*Chaplin et al., 2021a*; *Chaplin et al., 2021b*; *Zhao et al., 2020*). The V(D)J recombination-specific 'two-synapses' apparatus appears as a double-edged sword backup system to avoid genomic insta- bility. Indeed, RAG2/Xlf double mutant mice develop typical NHEJ-deficient pro-B cell lymphomas when crossed onto a *Trp53-/-* background (*Lescale et al., 2016a*).

One question remains as to the contribution of X4 in this model given that classical KO strategies do not allow to directly assess the function of X4 since its absence results *de facto* in a complete Lig4 deficiency. Reminiscent to the Xlf-deficient condition, human patients identified with *X4* mutations present DNA repair defect hallmarks, including dwarfism, microcephaly, increased cellular sensitivity to radiomimetic agents, but, strikingly, no immunodeficiency (*de Villartay, 2015*; *Saito et al., 2016*). The absence of noticeable immunophenotype argues for a dispensable role of X4 during V(D)J recom- bination, outside its general Lig4 stabilization function. It also interrogates the positioning of X4 within

the 'two-synapses' model knowing its structural and functional relationship with Xlf (*Callebaut et al., 2006*).

XRCC4 is a keystone factor, playing two independent roles. First, X4 stabilizes Lig4 through its C-terminal coiled-coil domain (*Grawunder et al., 1997*). Indeed, Lig4 expression is abrogated in *Xrcc4* Knock Out (KO) models both in vivo and in vitro (*Gao et al., 1998*). In addition, X4 and Xlf homodimers interact through their N-terminal globular head to form long polymeric 'filaments' (*Reid et al., 2015*; *Ropars et al., 2011*). X4-Xlf filaments generate a 'DNA repair synapse' to tether broken DNA ends (*Brouwer et al., 2016*; *Reid et al., 2015*). DNA-end synapsis is a central issue during NHEJ, and the recent development of in vitro single-molecule technologies has highlighted the dynamic formation of DNA end-to-end synapses (flexible/long range for DNA end tethering and close/short range for DNA ligation) in addition to the Xlf-X4 filament, in which the various core NHEJ DNA repair factors (Ku70/80, DNA-PKcs, Xlf, X4/L4, and PAXX) participate to various degrees, in particular the association of Xlf with both X4 and Ku (for a recent review, see *Zhao et al., 2020*). Several studies recently reported on the details of the structural assembly of these complexes using cryo-EM, thus improving our understanding on the composition of these complexes as well as the dynamics of the transition between various states during NHEJ-mediated DNA repair (*Chaplin et al., 2021a*; *Chaplin et al., 2021b*; *Zhao et al., 2020*). These studies support in particular the interaction of the L4X4 complex with that of Ku70/80 previously proposed by *Costantini et al., 2007*. The V(D)J recombination phenotype of *Nhej1* KO mice would argue that the RAG2-mediated DNA tethering is also redundant with these DNA end-to-end synapses. Nevertheless, the intimate nature of DNA end joining during V(D)J recombination may not always strictly coincide with what we know for the repair of genotoxic DNA breaks, precisely because of the existence of the 'two-synapses' mechanism.

Since X4 is compulsory for Lig4 stabilization, *Xrcc4-/-* mice phenocopy *Lig4-/-* condition, with an embryonic lethality and SCID phenotype (*Gao et al., 1998*). The embryonic lethality is rescued on *Trp53-/-* background, but the SCID resulting from a complete block of lymphocyte development remains (*Gao et al., 2000*). Furthermore, *Xrcc4-/-Trp53-/-* DKO mice develop Pro-B cell lymphomas (*Chen et al., 2016*; *Gao et al., 2000*). To avoid disrupting the critical Lig4 stabilization function of X4, we engineered a *Xrcc4 knock-in* (KI) mouse model harboring the M61R missense mutation that abrogates X4-Xlf interaction (*Ropars et al., 2011*), while keeping the Lig4 interaction domain unperturbed. *Xrcc4$^{M61R}$* mice are viable attesting for the stabilization of functional Lig4, thus allowing the study of lymphocyte development. Introduced on *Atm-/-*, *Paxx-/-*, and *Nhej1-/-* backgrounds, the *Xrcc4$^{M61R}$* mutation allows to address the 'two-synapses' DNA repair model in V(D)J recombination, and to expand the picture of NHEJ apparatus in brain and lymphocyte development.

## Results
### Generation of X$^{M61R}$ mice
X4 and Xlf interact through a hydrophobic interface, which is disrupted by the X4$^{M61R}$ substitution, thus resulting in the loss of X4-Xlf filament (*Ropars et al., 2011*). We developed an *Xrcc4$^{M61R}$* KI mouse model through CRISPR/Cas9 (*Figure 1A*). Homozygous *Xrcc4$^{M61R/M61R}$* mice (*Xrcc4$^{M61R}$* mice) were viable arguing against any harmful impact of the M61R substitution during embryonic development, as opposed to the embryonic lethality of *Xrcc4* KOs.

### X4$^{M61R}$ stabilizes Lig4
RT-PCR analysis of X4 mRNA expression unveiled two isoforms in M61R mutation-bearing cells (*Figure 1B*). The expression of the full-length transcript carrying the five mutations was reduced (*Figure 1B*, upper band) and was accompanied by a shorter isoform (*Figure 1B*, lower band). Sequencing this alternative transcript revealed the out-of-frame splicing-out of exon 3 (*Figure 1—figure supplement 1*). The same X4 transcript missing exon 3 was described in a previous *Xrcc4* KO mouse model (*Gao et al., 2000*; *Gao et al., 1998*). A subsequent attempt to generate *Xrcc4$^{M61R}$* KI mouse model using an ssODN template lacking the four silent mutations resulted in the same pattern of expression of the M61R allele (data not shown). At the protein level, X4 expression was barely detectable in spleen, brain, thymus, and mouse embryonic fibroblasts (MEFs) from *Xrcc4$^{M61R}$* mice as opposed to their homozygous WT counterparts (*Figure 1C and D*). In summary, *Xrcc4$^{M61R}$* mice express two X4 transcripts, one lacking exon 3 and resulting in a complete loss of function, and a

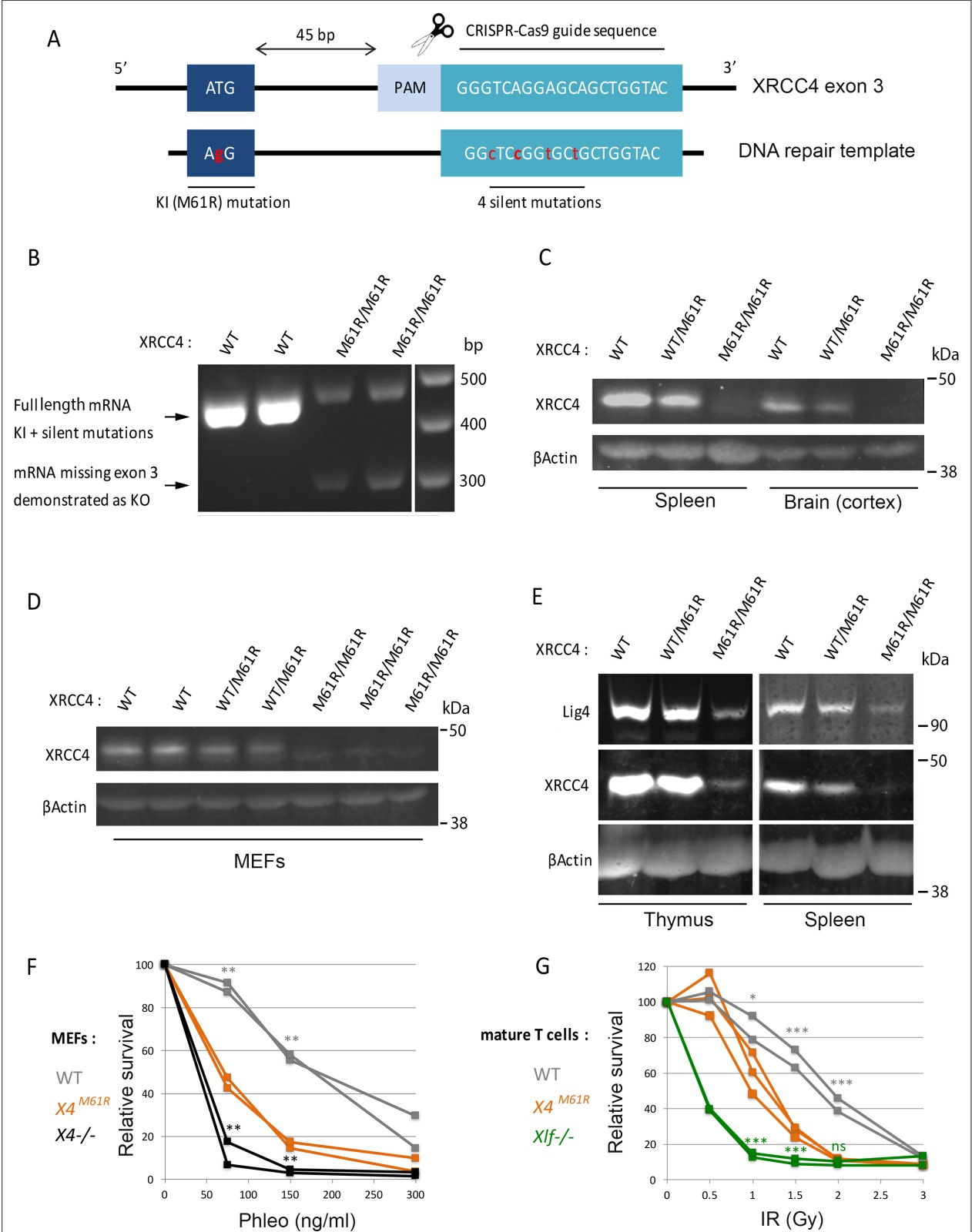

**Figure 1.** X4 and Lig4 expression and impaired non-homologous end-joining (NHEJ) in *Xrcc4*[M61R] mice. (**A**) Schematic representation of CRISPR/Cas9-driven homologous recombination strategy to generated *Xrcc4*[M61R] knock-in mouse model. (**B**) 5'UTR to 3'UTR *Xrcc4* RT-PCR in mouse thymocyte extracts from four littermates. The lower transcript represents the splicing out of exon 3 as described (*Gao et al., 1998*). (**C**) WB analysis from mouse whole tissue extracts. (**D**) Western blot (WB) analysis from mouse embryonic fibroblast (MEF) extracts. (**E**) WB analysis from mouse whole tissue extracts.

*Figure 1 continued on next page*

*Figure 1 continued*

WB were performed at least three times with two animals per genotypes. (**F**) MEFs' sensitivity to DNA double-strand break (DSB)-inducing agent phleomycin. Statistical tests for *Xrcc4*$^{M61R}$ vs. WT (gray *) and *Xrcc4*$^{M61R}$ vs. *Xrcc4*-/- (black *). (**G**) Relative survival of CD3/CD28-activated mature T cells following irradiation and 4 hr recovery. Statistical analysis for *Xrcc4*$^{M61R}$ vs. WT (gray *) and *Xrcc4*$^{M61R}$ vs. *Nhej1*-/- (green *).

The online version of this article includes the following figure supplement(s) for figure 1:

**Figure supplement 1.** Nucleotide sequence of two Xrcc4 transcripts in *Xrcc4*$^{M61R}$ mice.

second, full length, harboring the M61R missense variant encoding a weakly expressed X4 protein lacking its ability to interact with Xlf. Of note, Lig4 protein was readily detectable by WB in protein lysates from *Xrcc4*$^{M61R}$ mice spleen and thymus, attesting for its stabilization by the X4$^{M61R}$ protein (*Figure 1E*).

We conclude that the *Xrcc4*$^{M61R}$ mouse model represents a hypomorphic condition in which the X4$^{M61R}$ protein, although weakly expressed, retains the capacity to stabilize Lig4 and thus ensures proper embryonic development with viable animals at birth.

## DSB repair defect in X4$^{M61R}$ cells

MEFs from *Xrcc4*$^{M61R}$ mice presented a statistically significant decrease in cell viability compared to WT cells upon phleomycin, a DSB inducer, treatment, attesting for a profound DNA repair defect (*Figure 1F*). Likewise, splenic T cells from *Xrcc4*$^{M61R}$ mice showed an increased IR sensitivity with an almost complete loss of viability at 2 Gy (*Figure 1G*). These experiments demonstrate that the X4$^{M61R}$ mutant protein, which preserves the stabilization of Lig4, is nevertheless impaired in its capacity to achieve full repair of DSB introduced by random genotoxic agents.

## Lymphocyte development in *Xrcc4*$^{M61R}$ mice phenocopies that of *Nhej1*-/- mice

We analyzed the impact of X4$^{M61R}$ substitution on T cell development in the thymus. Thymocyte counts were significantly reduced in *Xrcc4*$^{M61R}$ thymus (*Figure 2A*), but remained in the range of *Nhej1*-/- thymus cellularity (respectively [*10$^6$] 26.8 ± 2.6 SEM vs. 115 ± 15 SEM in WT, p<0.0001, vs. 12.5 ± 2.6 SEM in *Nhej1*-/-, ns). The thymocyte maturation program, identified through the various CD4/CD8 stages, was similar in *Xrcc4*$^{M61R}$ and WT littermates (*Figure 2B*), yet with a mild accumulation of CD44-CD25+ CD28- DN3A *Xrcc4*$^{M61R}$ thymocytes (respectively 81.8% ± 0.9 SEM vs. 76.1% ± 0.7 SEM, p<0.0001) (*Figure 2C*), a trait shared by thymocytes from *Nhej1*-/- mice (81.1% ± 1.4 SEM) as described (*Roch et al., 2019*). Purified *Xrcc4*$^{M61R}$ thymocytes presented an increased apoptosis (7AAD+/AnnV+), similar to that of *Nhej1*-/- thymocytes (respectively 47.4% ± 1.7 SEM vs. 33.2% ± 1.3 SEM for WT, p=0.0003, vs. 59.5% ± 1.6 SEM, for *Nhej1*-/-, ns) (*Figure 2D and E*). This was accompanied by the expression of TP53 target genes *Cdkn1a*, *Bbc3*, and *Bax*, in both *Xrcc4*$^{M61R}$ and *Nhej1*-/- thymocytes (*Figure 2F*). Lastly, TCRα repertoire was skewed in *Xrcc4*$^{M61R}$ thymocytes (*Figure 2G*), with the decreased usage of distal VαJα rearrangements, a hallmark of reduced thymocyte viability or suboptimal V(D)J recombination.

The proportion of mature B lymphocytes in the bone marrow (BM) was not significantly altered in *Xrcc4*$^{M61R}$ mice (*Figure 2H, I*), while significantly decreased in *Nhej1*-/- mice (respectively 21.7% ± 1.8 SEM vs. 26.0% ± 1.1 SEM, ns, vs. 13.3% ± 0.6 SEM, p=.008). However, the proportions of immature B cells were decreased in *Xrcc4*$^{M61R}$ mice (*Figure 2J*), close to the already described decline of *Nhej1*-/- immature B cells (respectively 11.3% ± 3.3 SEM vs. 19.3% ± 2.6 SEM, p=0.0002, vs. 8.7% ± 2.5 SEM, ns).

These results indicate that B- and T-lymphocyte development is not severely affected in *Xrcc4*$^{M61R}$ mice, despite the reduced expression of X4$^{M61R}$. The minor accumulation of DN3A thymocytes and immature B cells suggests a modest decreased V(D)J recombination efficiency, supported by the bias in TCRα usage. This is in sharp contrast with *Xrcc4*-/- and *Lig4*-/- mice, which present a complete arrest of lymphocyte maturation and a SCID phenotype. We conclude that when Lig4 is stabilized, the contribution of X4 through Xlf/X4 interaction, disrupted in the context of X4$^{M61R}$, is not critical for V(D)J recombination in vivo.

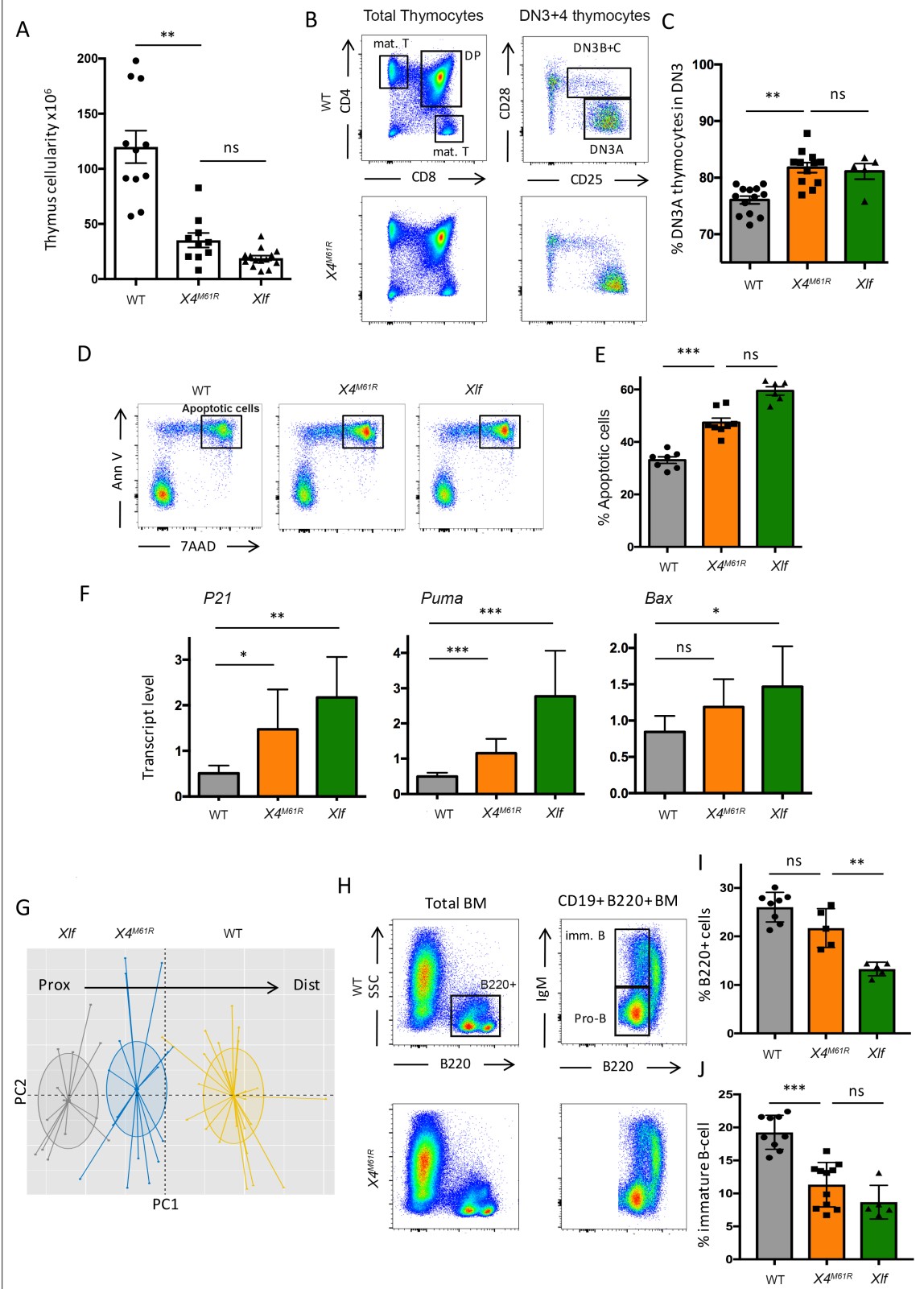

**Figure 2.** *Xrcc4*[M61R] mice exhibit modest T cell development defect in the thymus phenocopying *Nhej1-/-* mice. (**A**) Thymus cellularity of WT, *Xrcc4*[M61R], and *Nhej1-/-* (Xlf) 6–9 -week-old mice. (**B**) Immunostaining of thymus cellular populations. Various relevant populations are highlighted in black gates. (**C**) Quantification of DN3A thymocyte subpopulation (DN3 CD28-CD25+) gated from total DN3 thymocytes (CD4-CD8-CD44-CD25+). (**D**) Thymocyte apoptosis analysis after 20 hr of culture. AnnexinV+/7AAD+ apoptotic cells are highlighted in black gates. (**E**) Quantification of thymocyte apoptosis

*Figure 2 continued on next page*

*Figure 2 continued*

(AnnexinV+/7AAD+ cells) after 20 hr of culture. (**F**) Quantitative RT-PCR analysis of TP53 target genes *Cdkn1a* (encoding P21), *Bbc3* (encoding PUMA), and *Bax* in thymocyte extracts. There is no statistical difference between *Xrcc4^M61R^* and *Nhej1-/-* in the expression of these TP53 target genes. (**G**) Illustrative representation of principal component analysis/unsupervised hierarchical clustering (PCA/HC) analysis of mTRAJ-mTRAV combinations in thymus according to PROMIDISα. (**H**) Immunostaining of bone marrow cellular populations. V(D)J recombination-dependent developmental stages are highlighted within red rectangles. Various relevant populations are highlighted in black gates. Quantification of B-cell population (B220+) (**I**) and immature B-cell population (CD19+ B220+ IgM+) (**J**) from total bone marrow.

## PAXX and ATM are compensatory factors for X4 in lymphoid development

We previously proposed a model of 'double DNA repair synapse' during V(D)J recombination according to which the X4-Xlf 'filament' and the RAG1/2 PCC, together with ATM and PAXX, provide two complementary means of DNA end synapsis (*Abramowski et al., 2018*; *Betermier et al., 2020*; *Lescale et al., 2016a*). To address the role of X4 in this model, we crossed *Xrcc4^M61R^* onto *Paxx-/-* and *Atm-/-* mice (hereafter denoted as *Xrcc4^M61R^-Paxx* and *Xrcc4^M61R^-Atm*). Both double mutant mice were viable but exhibited severe growth retardation and facial dysmorphia as a consequence of either the X4^M61R^ mutation and/or the overall reduction in X4/L4 expression (*Figure 3—figure supplement 1*).

The thymus and spleen cellularity was severely reduced in both *X4^M61R^-Paxx* and *Xrcc4^M61R^-Atm* mice (*Figure 3A and B*) as compared to their *Xrcc4^M61R^* single-mutant littermates (respectively [*10^6^] 1.1 ± 0.2 SEM and 0.56 ± 0.15 SEM, p<0.0001, vs. 26.8 ± 2.6 SEM in thymus and 6.8 ± 1.2 SEM and 4.1 ± 1.0 SEM, p<0.0001, vs. 40.0 ± 3.8 SEM in spleen), and previously described Atm and Paxx single KO conditions (*Abramowski et al., 2018*). B cells expressing IgM were undetectable in BM from *Xrcc4^M61R^-Atm* mice and to a lesser extent in *Xrcc4^M61R^-Paxx* mice (*Figure 3C*). B220+ B cells were strongly reduced compared to *Xrcc4^M61R^* or WT littermates in both settings (respectively 10.9% ± 1.4 SEM and 7.2% ± 0.96 SEM, p=0.001, vs. 21.7 ± 1.79 SEM in *Xrcc4^M61R^* and 26.3 ± 1.08 SEM in WT) (*Figure 3D*). Likewise, *Xrcc4^M61R^-Paxx* and *X4^M61R^-Atm* mice presented with a substantial accumulation of DN3A thymocytes (93.4% ± 0.8 SEM and 98.2% ± 0.3 SEM, p=0.0001, respectively, vs. 76.1 ± 0.7 SEM in WT), which was significantly greater than the one observed for *Xrcc4^M61R^* (81.77 ± 1.4 SEM) or *Nhej1-/-* (81.1 ± 1.4 SEM) animals (*Figure 3E* and *Abramowski et al., 2018*). The immune phenotype of *Xrcc4^M61R^-Paxx* and *X4^M61R^-Atm* mice is highly reminiscent of the SCID condition experienced by *Nhej1-Paxx*, *Nhej1-Atm*, and *Nhej1-Rag2^c/c^* DKO mice (*Abramowski et al., 2018*; *Lescale et al., 2016a*; *Musilli et al., 2020*; *Zha et al., 2011*) and suggested a possible impaired V(D)J recombination in these double mutants. Indeed, while Dβ₂-Jβ₂ (upper panel) and Vβ₁₀-Dβ₂Jβ₂ (lower panel) rearrangements of the *Tcrb* locus were present in thymocytes from *Xrcc4^M61R^*, *Atm-/-*, and WT mice (Figure 3F and G), as also previously observed in *Paxx-/-* mice (*Abramowski et al., 2018*), they were barely detectable in DNA from *Xrcc4^M61R^-Paxx* and *Xrcc4^M61R^-Atm* thymus.

We conclude that *Xrcc4^M61R^* crossed on *Paxx-/-* or *Atm-/-* phenocopy the SCID phenotype of *Nhej1-Paxx* and *Nhej1-Atm* DKO conditions owing to impaired V(D)J recombination. Therefore, aside from its essential role in Lig4 stabilization, X4 does participate in DNA coding-end tethering through its interaction with Xlf during V(D)J recombination, and PAXX/ATM are compensatory factors for this later function in vivo, as they are for Xlf. Alternatively, one may consider that the Xlf function being impaired in the context of X4^M61R^, PAXX/ATM complements this defect as already established.

## Impaired brain development and immune deficiency in *Xrcc4^M61R^ Nhej1-/-* mice

To further assess the function of X4^M61R^ in vivo, we introduced *Xrcc4^M61R^* on *Nhej1-/-* background. No *Xrcc4^M61R^-Nhej1-/-* double homozygous mice (hereafter named as *Xrcc4^M61R^-Nhej1*) could be recovered out of 166 newborns obtained from various genotype combinations, when 23 were expected (*Figure 4A*), testifying the lethality of *Xrcc4^M61R^-Nhej1* embryos. Nevertheless, lived *Xrcc4^M61R^-Nhej1* fetuses could be recovered at E15.5 and E18.5. They were smaller than their *Xrcc4^M61R^-Nhej1+/-* littermates, confirming abnormal development (*Figure 4B* and data not shown). This is reminiscent of the late embryonic lethality of *Lig4-/-*, *Xrcc4-/-*, and *Nhej1-Paxx* DKO mice (*Abramowski et al., 2018*; *Balmus et al., 2016*; *Frank et al., 1998*; *Gao et al., 1998*; *Liu et al., 2017*). Massive neuronal apoptosis was detected in E15.5 brain slices through cleaved-caspase 3 (CC3) staining in upper layers of dorsal telencephalon corresponding to the intermediate zone (IZ) and the cortical plate (CP) in

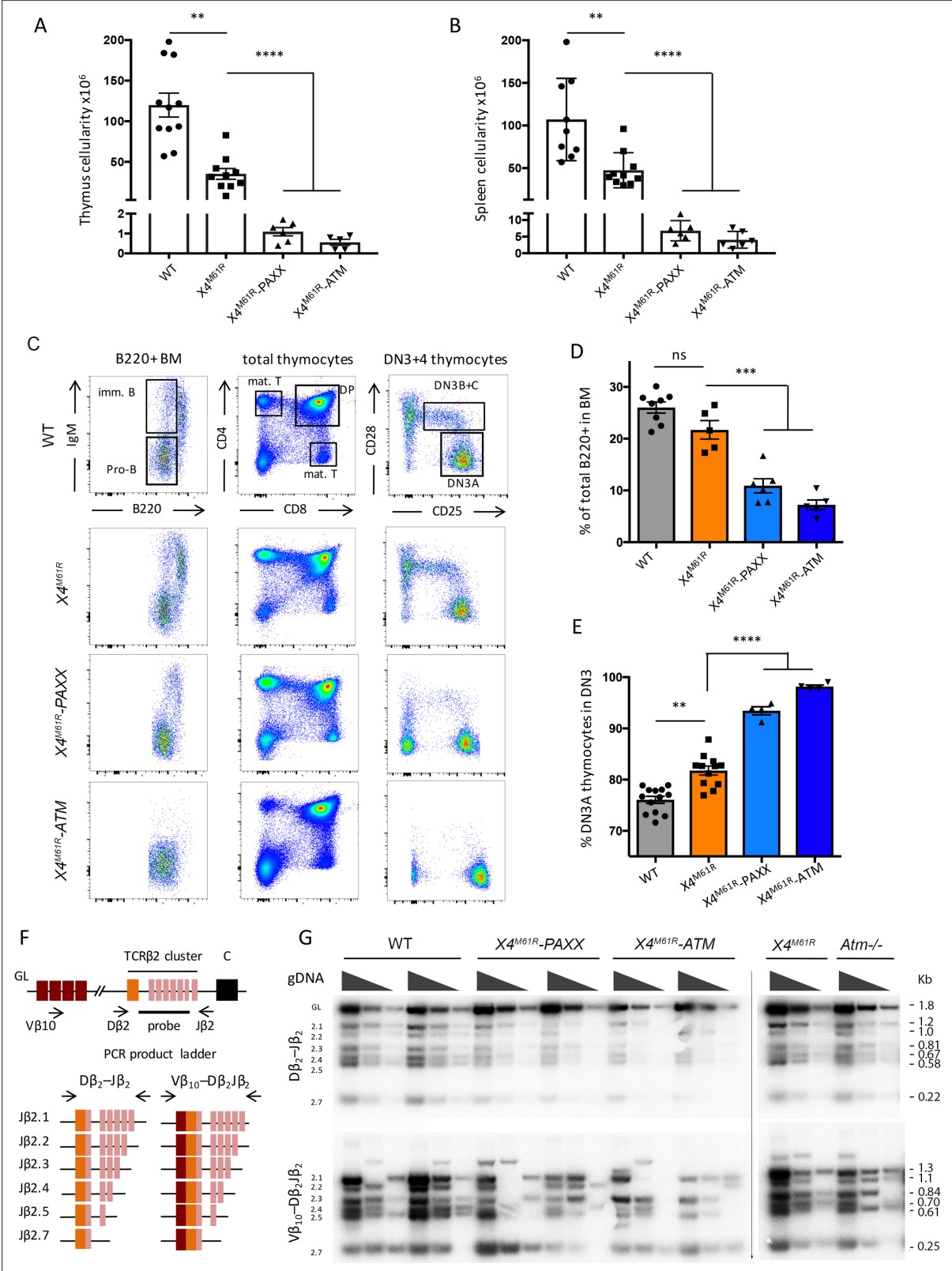

**Figure 3.** PAXX and ATM are compensatory factors of X4$^{M61R}$ in immune development. (**A, B**) Thymus and spleen cellularity of WT, *Xrcc4*$^{M61R}$, *Xrcc4*$^{M61R}$–*Paxx*, and *Xrcc4*$^{M61R}$–*Atm* 6–9 -week-old mice. (**C**) Immunostaining of BM and thymus cellular populations. Various relevant populations are highlighted in black gates. (**D**) Quantification of B220+ subpopulation of total BM. (**E**) Quantification of DN3A thymocyte subpopulation (DN3 CD28-CD25+) of total DN3 thymocytes (CD4-CD8-CD44-CD25+). (**F**) Schematic representation of PCR strategy to analyze *Tcrb* rearrangement according to ***Abramowski***

*Figure 3 continued on next page*

*Figure 3 continued*

**et al., 2018**. (**G**) Autoradiogram of ladder of productive Dβ$_2$-Jβ$_2$ and Vβ$_{10}$-Dβ$_2$Jβ$_2$ semi-quantitative PCR products revealed by the TCR-Jβ probe. Genomic DNA dilutions are represented with triangles. Germline allele configuration (GL) is revealed by the upper band of Dβ$_2$-Jβ$_2$ PCR. Southern were performed twice with two animals per genotype.

The online version of this article includes the following figure supplement(s) for figure 3:

**Figure supplement 1.** Growth retardations of *Xrcc4*$^{M61R}$-*Paxx* and *Xrcc4*$^{M61R}$-*Atm* mice.

*Xrcc4*$^{M61R}$-*Nhej1* mice (**Figure 4C–E**). The IZ contains post-mitotic neurons migrating to populate the CP, while neural progenitors proliferate in lower layers: the ventricular zone (VZ) and sub-ventricular zone (SVZ). Thus, *Xrcc4*$^{M61R}$-*Nhej1* embryonic lethality recapitulates, although to a lesser extent, the massive post-mitotic neuron apoptosis observed in *Nhej1-Paxx* DKO and *Xrcc4-/-* brains (**Figure 4C and E**), which causes the previously described embryonic lethality of these conditions. In contrast, *Nhej1-/-* and *Xrcc4*$^{M61R}$ single mutants showed modest increases of CC3 staining (**Figure 4D and E**), while we showed previously that *Paxx-/-* are indistinguishable from WT (**Abramowski et al., 2018**).

We next analyzed the adaptive immune system development of *Xrcc4*$^{M61R}$-*Nhej1* fetuses. B-lymphocyte development begins at E17.5 in the fetal liver (FL), with the rearrangement of the *Igh* locus and the intracellular expression of Ig-µH chains in CD19+ B220+ CD43+ Pro B cells, which is used as a proxy to evaluate *Igh* rearrangement completion (**Kajikhina et al., 2016**). *Xrcc4*$^{M61R}$-*Nhej1* Pro-B cells failed to express intracellular µH chain when compared to WT (0.90% ± 0.09 SEM, p=0.008), which contrasted with the slight reduction observed in *Xrcc4*$^{M61R}$ and *Nhej1-/-* mice (respectively 16.6% ± 1.2 SEM and 19.1% ± 2.2 SEM vs. 32.8% ± 1.3 SEM, p=0.0016) (**Figure 5A and B**), arguing for a severe block in B cell maturation at Pro-B cell stage in *Xrcc4*$^{M61R}$-*Nhej1* mice. Fetal thymocytes development of *Xrcc4*$^{M61R}$ and *Nhej1-/-* mice recapitulated young adult thymus phenotype with a modest block at DN3A stage compared to WT (respectively 83.3% ± 2.7 SEM and 86.2 ± 2.2 SEM vs. 70.3% ± 1.1 SEM, p=0.0004). In contrast, *Xrcc4*$^{M61R}$-*Nhej1* thymocyte development was severely compromised from the DN3A stage on when compared to *Xrcc4*$^{M61R}$ and *Nhej1-/-* (99.4% ± 0.13 SEM, p=0.0007), and DP thymocytes or mature T cells were virtually absent from these thymii (**Figure 5A and C**). Dβ$_2$-Jβ$_2$ and Vβ$_{10}$-Dβ$_2$Jβ$_2$ *Tcrb* genes rearrangements, which begin at around E13 (**Ramond et al., 2014**), were undetectable in *Xrcc4*$^{M61R}$-*Nhej1* thymus, while readily observed in WT, *Xrcc4*$^{M61R}$, and *Nhej1-/-* mice (**Figure 5D and E**), further attesting for an acute V(D)J recombination defect in *Xrcc4*$^{M61R}$-*Nhej1* cells.

Altogether, *Xrcc4*$^{M61R}$-*Nhej1* embryonic development phenocopy *Nhej1-Paxx* DKO, with a profound defect in V(D)J recombination and the apoptosis of post-mitotic neurons leading to embryonic lethality.

## Discussion

We created an *Xrcc4* separation of function allele in mice, which harbor the M61R substitution previously described to abolish X4-Xlf interaction (**Ropars et al., 2011**). The X4$^{M61R}$ protein, although weakly expressed, preserves the stabilization of Lig4 but severely compromises the NHEJ DNA repair function in affected mice. Nevertheless, *Xrcc4*$^{M61R}$ mice are viable, arguing that the embryonic lethality of *Xrcc4* KO mice is most likely the result of the *de facto* absence of Lig4 in this setting, thus phenocopying *Lig4* KO condition. Interestingly, this also indicates that the formation of the X4-Xlf filament, abrogated by the M61R mutation, is dispensable for post-mitotic neuron viability, the loss of which causes lethality in *Xrcc4* and *Lig4* KO settings.

Immune development of *Xrcc4*$^{M61R}$ mice phenocopied that of *Nhej1-/-* animals, with a slight decreased thymic cellularity, an increased thymocyte apoptosis, and skewed TCRα repertoire, which we previously linked to a suboptimal *Tcra* rearrangement efficiency (**Berland et al., 2019**; **Roch et al., 2019**; **Vera et al., 2013**). In addition to their TCRα repertoire bias, *Xrcc4*$^{M61R}$ thymocytes also faced a developmental delay at DN3A both in fetal and adult thymus like *Nhej1* KO condition, further attesting for a suboptimal V(D)J recombination activity in these mice. Therefore, although X4 is required for V(D)J recombination through Lig4 stabilization, it is not mandatory for coding-ends tethering during V(D)J recombination. These results confirm that X4 exerts a Lig4-independent function through its interaction with Xlf that is decisive for the repair of genotoxic-induced DSBs but compensated for by other DNA repair factors, and possibly RAG2 itself as shown for Xlf (**Lescale et al., 2016a**), during V(D)J recombination. *Xrcc4*$^{M61R}$–*Paxx* and *Xrcc4*$^{M61R}$–*Atm* double homozygous mice

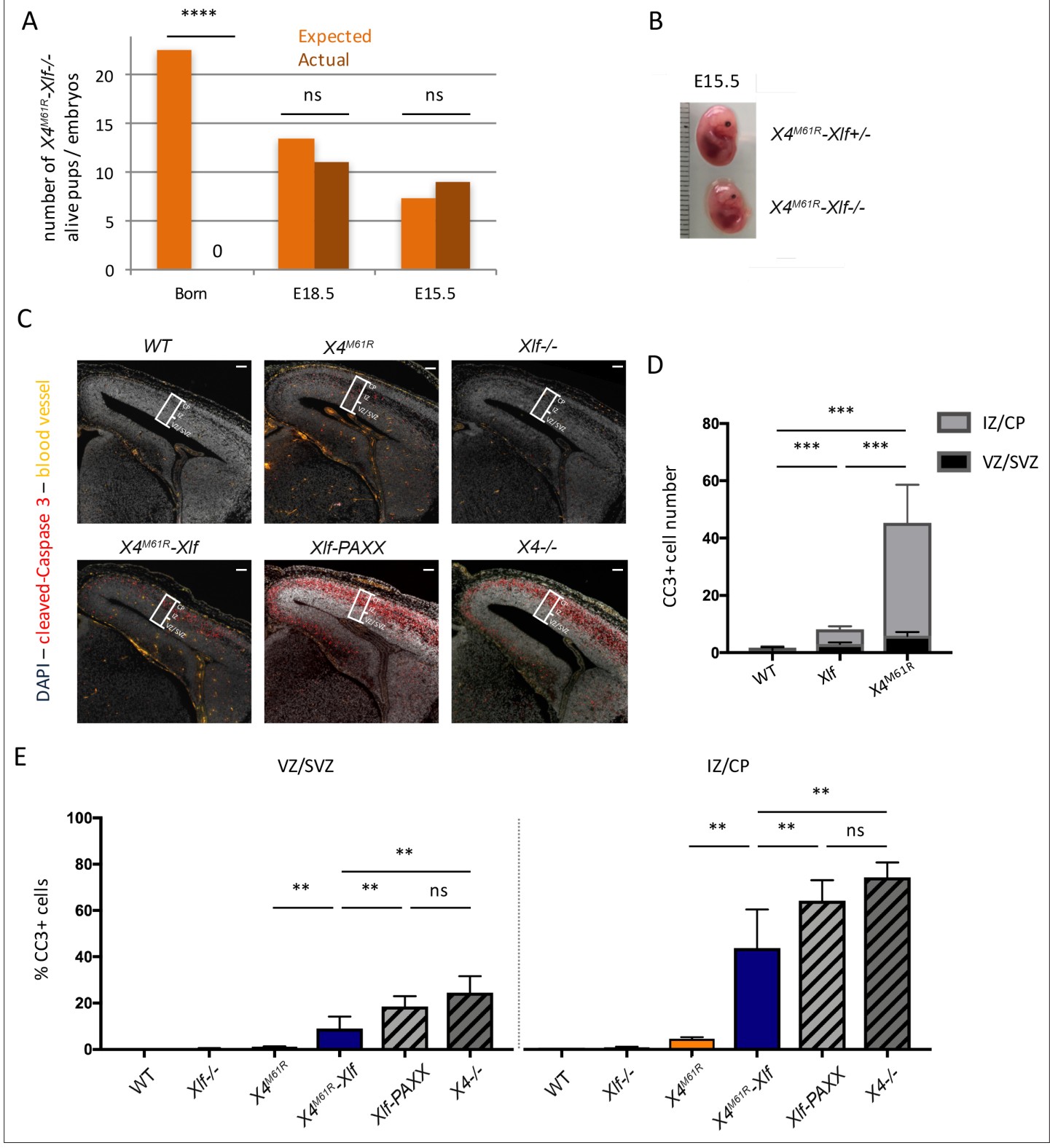

**Figure 4.** Embryonic lethality and neuronal apoptosis of *Xrcc4*[M61R]-*Nhej1* embryos. (**A**) Impaired development of *Xrcc4*[M61R]-*Nhej1* mice. Actual and expected numbers of *Xrcc4*[M61R]-*Nhej1* alive pups, E18.5 and E15.5 embryos. (**B**) Pictures of *Xrcc4*[M61R] and *Xrcc4*[M61R]-*Nhej1* E15.5 embryos. (**C**) Cleaved-caspase 3 (CC3, red) immunostaining of E15.5 brain slices. Scarce apoptotic cells are detected in the whole dorsal telencephalon from *Nhej1-/-* and *Xrcc4*[M61R] embryos. By contrast, massive neuronal apoptosis is observed in the upper layers of the developing cortex of *Xrcc4*[M61R]-*Nhej1*, *Xrcc4-/-*, and *Nhej1-Paxx* DKO embryos. The white boxes indicate the location of the standard window (100 µm wide) spanning from the ventricular to pial surface

*Figure 4 continued on next page*

*Figure 4 continued*

used to quantify pyknotic nuclei in the dorsal telencephalons (see below). Scale bar: 100 μm. (**D**) Number of CC3-positive cells in the VZ/SVZ (black)—containing the cycling neural progenitors—and intermediate zone (IZ)/cortical plate (CP) (gray)—containing post-mitotic neurons—of the whole dorsal telencephalons from WT, *Nhej1-/-*, and *Xrcc4$^{M61R}$* embryos. (**E**) Percentage of apoptotic (pyknotic) nuclei in the ventricular zone (VZ)/sub-ventricular zone (SVZ) (left panel) and the IZ/CP (right panel) found in the standard windows shown in (**C**). Data were obtained from both hemispheres of three embryos per condition.

were severely immunocompromised, owing to aborted V(D)J recombination, as noticed in *Nhej1-Paxx* and *Nhej1-Atm* DKO. We conclude that PAXX and ATM are compensatory factors for X4$^{M61R}$ during V(D)J recombination. This result further supports our previously proposed model of 'double DNA repair synapse' in V(D)J recombination, mediated by the X4-Xlf and RAG2-PAXX-ATM axes, respectively (*Abramowski et al., 2018*; *Betermier et al., 2020*; *Lescale et al., 2016a*). In addition to their defect in V(D)J recombination, *Xrcc4$^{M61R}$-Nhej1* double homozygous mice were embryonic lethal owing to massive post-mitotic neuron apoptosis, recapitulating *Xrcc4-/-* setting. This observation may be the result of the lower expression of the X4$^{M61R}$ protein, which could have additive/synthetic effects with the loss of Xlf interaction during NHEJ. Indeed, *Nhej1* deficiency causes synthetic lethality with several other NHEJ-deficient conditions, such as *Prkdc-/-*, *Mri-/-*, *H2ax-/-*, and *Paxx-/-* (*Abramowski et al., 2018*; *Hung et al., 2018*; *Xing et al., 2017*; *Zha et al., 2011*). Therefore, Xlf can rescue very different NHEJ defects, through an unknown mechanism, perhaps in relation with recovery of replication fork stalling as shown in the *Nhej1-H2ax* DKO mice (*Chen et al., 2019*).

Altogether, the *Xrcc4$^{M61R}$* separation of function allele highlights for the first time at least two independent roles of X4 in NHEJ and establishes that X4 is not mandatory for coding ends tethering during V(D)J recombination. This study also unravels novel interplays between X4, PAXX, ATM, and Xlf during development of the brain and the immune system.

Several deleterious mutations in the *XRCC4* gene have been reported in humans (see *de Villartay, 2015* for review), most of which are associated with microcephalic primordial dwarfism (MPD), gonadal failure, early-onset metabolic syndrome, and cardiomyopathies. The DNA repair deficiency in these patients is manifest when tested in vitro retrospectively. Most surprisingly however, they do not present noticeable signs of immune dysfunction. Altogether, the clinical presentation of these patients was not evocative of an impaired NHEJ given the known impact of its deficiency on the development of the adaptive immune system. Indeed, *X4* mutations were not identified in these patients through hypothesis-driven candidate gene sequencing but rather through unsupervised whole-exome sequencing. Our present study now provides some hints as to explain the absence of immunological features in *X4*-deficient patients; when Lig4 expression is spared to some extent by hypomorphic mutations, allowing birth, XRCC4 appears not critically required for the development of the adaptive immune system.

## Materials and methods
### Mice

*Nhej1-/-* (*Vera et al., 2013*), *Paxx-/-* (*Abramowski et al., 2018*), *Xrcc4-/-* (*Gao et al., 1998*), and *Atm-/-* (*Barlow et al., 1996*) mice were maintained in pathogen-free environment. All experiments were performed in compliance with the French Ministry of Agriculture's regulations for animal experiments (act 87847, 19 October 1987; modified in May 2001).

The *Xrcc4$^{M61R}$* allele was generated through CRISPR/cas9 ribonucleoprotein (RNP) complex microinjection in C57BL/6J mouse zygotes pronuclei as described (*Ucuncu et al., 2020*). *Xrcc4* exon 3-specific gRNA (5′-GTACCAGCTGCTCCTGACCC-3′) was designed using Crispor (http://crispor.tefor.net/oligonucleotide). The HR template was the ssODN ([Integrated DNA technologies, IDT] 5′-TCAAGGGAGAAATGCCGAGACTCCTTAGAAAAGAGGAACTTGTATGTACCAGCaGCACCGGAGCCTGGCACCAGTGCCTTTCTCAGCTCATCAATGTATTTTCCTTTCTCCCTAGCCATGTCATCAGCTTCTTGGGAAATCTCCAATTCAGAAACTATGGGAAAGATTAATTAAGGTGAT-3′; see *Figure 1—figure supplement 1*). F° chimera screen and *Xrcc4$^{M61R}$* mice genotyping were obtained upon FspBI digestion of genomic DNA amplified with primers F 5′-AGGAGACGGAGGAAAAAGAGATG-3′ and R 5′-TACCCTCACAGAAACACAACTCA-3′.

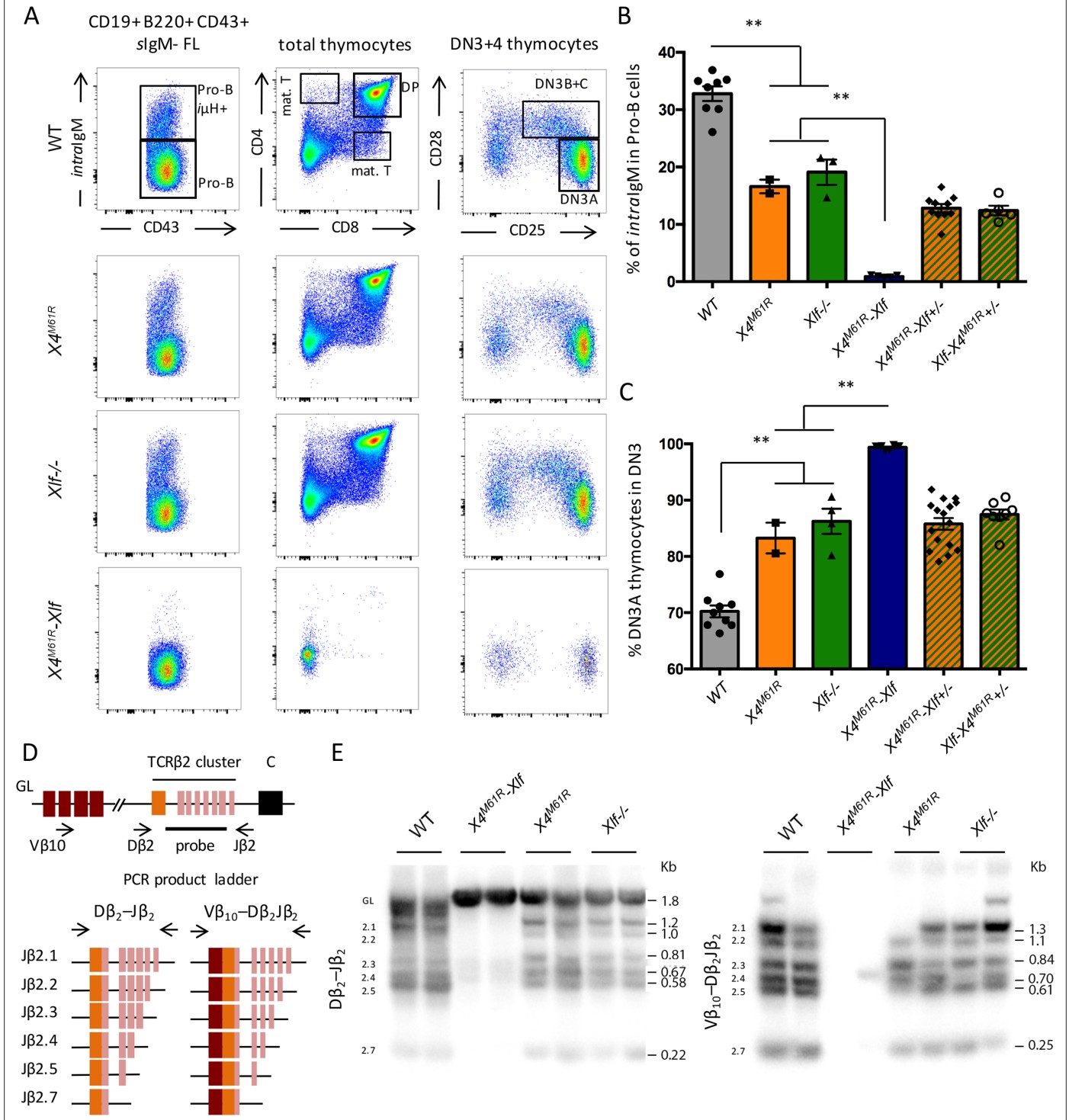

**Figure 5.** Complete V(D)J recombination defect in E18.5 *Xrcc4*[M61R]-*Xlf* embryos. (**A**) Immunostaining of E18.5 fetal CD19+ B220+ CD43+ hepatocytes and fetal thymocytes. Various relevant populations are highlighted in black gates. (**B**) Quantification of *intracellular* IgM expression in subpopulation of fetal Pro-B cells (CD19+ B220+ CD43+ *surface*IgM- hepatocytes). (**C**) Quantification of DN3A thymocyte subpopulation (DN3 CD28-CD25+) of total DN3 fetal thymocytes (CD4-CD8-CD44-CD25+). (**D**) Schematic representation of PCR strategy to analyze *Tcrb* rearrangement according to *Abramowski et al., 2018*. (**E**) Autoradiogram of ladder of productive Dβ2-Jβ2 and Vβ10-Dβ2Jβ2 direct PCR products revealed by the TCR-Jβ probe. Germline allele configuration (GL) is revealed by the upper band of Dβ2-Jβ2 PCR. Southern was performed once with two animal per genotype.

## Immunoblotting

Whole tissues or MEFs single-cell suspensions were washed in PBS. Cells were lysed in 50–200 µL of 50 mM Tris-HCl pH 7.5, 150 mM NaCl, 1% TritonX100, 1X phosphatase inhibitor cocktail 2 and 3 (ThermoFisher), and 1X protease inhibitor cocktail EDTA free (Roche) on ice for 45 min. Proteins migrated within Tris-Acrylamide gels and Tris-Glycine-SDS buffer and were transferred on PVDF membranes (Immobilon). Membranes were stained with the following antibodies: β-actin (mouse monoclonal AM4302, ThermoFisher), XRCC4 (goat monoclonal C-20, Santa Cruz), and DNA ligase IV (mouse monoclonal D-8, Santa Cruz). Membranes were revealed with appropriate secondary antibodies conjugated with infrared 700 or 800 dyes (LiCor). Immunoblotting was revealed by LiCor CLx analyzer and analyses were performed on Image Studio Lite software.

## X4 transcript analysis

Total RNAs were obtained from thymocytes with Pure Like RNA mini kit (Invitrogen) and used for RT-PCR (SuperScript II reverse transcriptase; Invitrogen). X4 transcripts were amplified with primers F1 5′-TTGGGCGCATCGGTTTATCT-3′ R1 5′-GCTGCTAAGTTGAAAGCCTG-3′ F2 5′-TGCTTCTGAACC CAACGTACC-3′ R2 5′- AGGTGCTCGTTTTTGGCTTG-3′. F1-R1 PCR amplifies full-length transcript from 5′UTR to 3′UTR (1150 bp) and F2-R2 amplifies exon 2 to exon 4 (400 bp).

## Flow cytometry analysis of cell populations

Cell phenotyping from 6- to 9- week-old mice was performed on thymus, spleen, and bone marrow using the following antibodies: CD4, CD8, CD25, CD28, CD44, CD69, B220, CD19, CD43, and IgM (all from Sony Biotechnology, using respectively PECy7, FITC, PerCPCy5.5, PE, BV510, APC, PE, PECy7, FITC, APC fluorophores). Intracellular IgM expression in E18.5 FL cells was performed as previously described (*Abramowski et al., 2018*) using CD19, B220, CD43, IgM for extracellular staining followed by cell fixation and permeabilization (Invitrogen) and intracellular IgM staining (all antibodies from Sony Biotechnology, using respectively PECy7, BV605, PE, FITC, and APC). Cells were recorded by FACS LSR-Fortessa X-20 and analyses were performed with FlowJo 10 software.

## Cellular sensitivity to DSBs-inducing agents and thymocyte survival assay

For phleomycin sensitivity assay, 5000 MEF cells were seeded in triplicates and cultivated with increasing doses (0–300 ng/mL) of phleomycin. Living cells were counted by flow analysis with FACS LSR-Fortessa X-20 after 6 days of culture. Radiation sensitivity of T lymphocytes was performed as described (*Abramowski et al., 2018*). Ex vivo thymocyte survival assay was performed as described (*Vera et al., 2013*).

## Quantitative real-time RT-PCR analysis

TaqMan PCR was performed on triplicates of 8 ng of reverse-transcribed RNA from freshly dissected total thymus as described (*Vera et al., 2013*).

## Analysis of thymic TCR-alpha repertoire

Comprehensive TCR-alpha repertoire analyses were performed by 5′ Rapid amplification of complementary DNA (cDNA) ends (5′RACE) PCR/NGS (switching mechanism at the 5′end of the RNA transcript, SMARTα) from total thymus RNA as described (*Abramowski et al., 2018*; *Roch et al., 2019*; *Vera et al., 2013*). Sequencing data were analyzed with LymAnalyzer (*Yu et al., 2016*) to retrieve unique CDR3 clonotypes and determine T cell receptor alpha variable (TRAV) and T cell receptor alpha junction (TRAJ) gene segments. Frequencies of TRAV and TRAJ usage were implemented in principal component analysis (PCA) and hierarchical clustering on principal components (HCPC) analyses using the *PCA*() and *HCPC*() functions of the FactomineR package, respectively (http://factominer.free.fr/; *Le et al., 2008*) and graphics were generated using the Factoextra R package (http://www.sthda.com/english/rpkgs/factoextra; *Kassambara, 2017*).

## TCRβ V(D)J recombination analysis

*Tcrb* rearrangements were analyzed by PCR on genomic DNA from total adult thymocytes or E18.5 fetal thymocytes as described (*Abramowski et al., 2018*).

## E15.5 brain sections immunohistochemistry

Neuronal apoptosis was analyzed as previously described (*Abramowski et al., 2018*). E15.5 fetal heads were fixed overnight at 4 °C by immersion in 4% paraformaldehyde and embedded in paraffin with a Tissu-tek processor (VIP, Leica). 5 µm coronal sections were then obtained using a microtome (Leica RM2125RT) and mounted onto glass slides for histological analyses. After paraffin removal and citrate treatment, the brain sections were permeabilized with 0.5% Triton X-100 in phosphate-buffered saline (PBS) for 15 min and incubated for 2 hr with 7.5% fetal bovine serum and 7.5% goat serum in PBS. The sections were incubated with rabbit anti-CC3 (Cell Signaling 9661) overnight at 4 °C. After washing, the sections were incubated with goat anti-rabbit Alexa Fluor 488 or 594 conjugated secondary antibody (ThermoFisher) for 1 hr. After washing, nuclear staining was achieved by incubation with 4′-6-diamidino-2-phenylindole (DAPI) to quantify apoptosis induction by the detection of pyknotic nuclei (*Roque et al., 2012*). Slides were mounted under Fluoromount (Southern Biotechnologies Associates). Tissues were examined under a fluorescence microscope (50i, Nikon, Japan) with a 10× (NA = 0.3) objective in three channels (appearing red, green, and gray) as separate files. These images were then stacked with Photoshop software (Adobe).

## Statistical analysis

Non-parametric Mann–Whitney test or two-tailed Fisher's exact test was performed with $\alpha$-risk = 0.05. p-Values were taken to be significant: *significant $0.05 \geq p > 0.01$; **very significant $0.01 \geq p > 0.001$; ***highly significant $0.001 \geq p > 0.0001$; ****highly significant $p \leq 0.0001$. Using G*Power software (https://www.psychologie.hhu.de/arbeitsgruppen/allgemeine-psychologie-und-arbeitspsychologie/gpower), we determined that a minimum of 47 newborns was necessary to conclude to an embryonic lethality of the $Xrcc4^{M61R}$-$Nhej1$-/- double KO with a power of 80% and a p-value < 0.05.

## Acknowledgements

This work was supported by institutional grant from INSERM and Université de Paris, state funding from the Agence National de la Recherche under 'Investissements d'avenir' program (ANR-10-IAHU-01) and grants from Institut National du Cancer (PLBIO 16-280) et Ligue Nationale contre le Cancer (Equipe Labellisée). BR received fellowships from INSERM, ARC, and LNCC.

# Additional information

## Funding

| Funder | Grant reference number | Author |
|---|---|---|
| Institut National de la Santé et de la Recherche Médicale | | Benoit Roch<br>Vincent Abramowski<br>Stefania Musilli<br>Jean-Pierre de Villartay |
| Agence Nationale de la Recherche | ANR-10-IAHU-01 | Benoit Roch<br>Vincent Abramowski<br>Stefania Musilli<br>Pierre David<br>Jean-Pierre de Villartay |
| Institut National Du Cancer | PLBIO 16-280 | Benoit Roch<br>Vincent Abramowski<br>Stefania Musilli<br>Jean-Baptiste Charbonnier<br>Isabelle Callebaut<br>Jean-Pierre de Villartay |
| Ligue Contre le Cancer | Equipe Labellisée | Benoit Roch<br>Vincent Abramowski<br>Stefania Musilli<br>Jean-Pierre de Villartay |

| Funder | Grant reference number | Author |
|--------|----------------------|--------|

The funders had no role in study design, data collection and interpretation, or the decision to submit the work for publication.

## Author contributions

Benoit Roch, Conceptualization, Formal analysis, Investigation, Methodology, Writing - original draft; Vincent Abramowski, Olivier Etienne, Stefania Musilli, Pierre David, Formal analysis, Investigation, Methodology; Jean-Baptiste Charbonnier, Isabelle Callebaut, Conceptualization; François D Boussin, Conceptualization, Data curation, Supervision, Validation, Writing - original draft; Jean-Pierre de Villartay, Conceptualization, Data curation, Formal analysis, Funding acquisition, Investigation, Project administration, Supervision, Validation, Writing - original draft

## Author ORCIDs

Jean-Pierre de Villartay http://orcid.org/0000-0001-5987-0463

## Ethics

All experiments were performed in compliance with the French Ministry of Agriculture's regulations for animal experiments (act 87847, 19 October 1987; modified in May 2001) after audit with "Comite' d'Ethique en Expe'rimentation Animale (CEEA) Paris Descartes" (Apafis #25432-2019041516286014 v6).

## Decision letter and Author response

Decision letter https://doi.org/10.7554/eLife.69353.sa1
Author response https://doi.org/10.7554/eLife.69353.sa2

# Additional files

## Supplementary files

• Transparent reporting form

## Data availability

The xl file with raw data used in PCA analysis (Fig. 2G) has been deposited on DRYAD https://data-dryad.org/stash/share/T4vNOWxXUUGanYkI2H--C39xErpsKK6kSynXqljpxsM.

The following dataset was generated:

| Author(s) | Year | Dataset title | Dataset URL | Database and Identifier |
|-----------|------|---------------|-------------|------------------------|
| Jean-Pierre D, Benoit R, Vincent A, Olivier E, Stefania M, Pierre D, Jean-Baptiste C, Isabelle C, François B | 2021 | Data from: A XRCC4 mutant mouse, a model for human X4 syndrome, reveals interplays with Xlf, PAXX, and ATM in lymphoid development | http://dx.doi.org/10.5061/dryad.547d7wm7x | Dryad Digital Repository, 10.5061/dryad.547d7wm7x |

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
