## [Decision Letter]

**Acceptance summary:**

The novel mouse model described in this manuscript helps to not only clarify the role of XRCC4 in VDJ recombination and, thereby, for immune cell development but also reveals new functional interactions between XRCC4 and XLF that regulate neuronal apoptosis. The separation of function point mutation studied here demonstrates that the primary role of XRCC4 during VDJ recombination is to stabilize Lig4. This distinction also provides a ready explanation for earlier observations that humans with hypomorphic mutations in XRCC4 are not clinically immunodeficient. These insights will be broad interest to the immune development and DNA repair research communities.

**Decision letter after peer review:**

Thank you for submitting your article "A XRCC4 mutant mouse, a model for human X4 syndrome, reveals interplays with Xlf, PAXX, and ATM in lymphoid development" for consideration by *eLife*. Your article has been reviewed by 2 peer reviewers, and the evaluation has been overseen by a Reviewing Editor and Satyajit Rath as the Senior Editor. The following individuals involved in review of your submission have agreed to reveal their identity: Michael Lieber (Reviewer #1); Karla Rodgers (Reviewer #2).

Essential revisions:

1) Provide a clearer and more detailed introduction to models of functions of XRCC4, XLF etc in end joining that highlights the fundamental question(s) being addressed via generation of the mouse model described in this manuscript.

2) Both reviewers list several inconsistencies in the figures, such as labeling of axes, percentages of cell populations, splenic cellularity in certain mutant mouse lines, scales in Figure 4, mean/standard deviations etc. these should be carefully checked and made uniform throughout the manuscript.

3) The protein levels of X4 and Lig4 in the double knock-out mice/embryos should be shown.

4) The Xlf-/- result is not shown in Figure 3E. Either the relevant citation needs to be included on this line, or the Xlf-/- result needs to be shown in Figure 3E.

The detailed comments from the reviewers are also appended below for providing context.

*Reviewer #1 (Recommendations for the authors):*

XRCC4 deficient mouse model exhibits late embryonic lethality and defective lymphogenesis. However, no clinically significant immunodeficiency was observed in human patients with XRCC4 (X4) hypomorphic mutations, even in those with no detectable X4 in the fibroblasts. The contradictory phenomena observed in mouse and human make the roles of X4 in lymphocyte development unclear. Roch et al., developed a unique X4 mutant mouse model and investigated the roles of X4 in V(D)J recombination. The authors first generated a viable X4M61R knock-in mouse model using CRISPR/Cas9 technique. The M61R mutation disrupts the interaction between X4 and XLF, but not the interaction of X4 and LIG4. The expression of X4 protein is severely inhibited in the homozygous X4 M61R/M61R (X4M61R) mice, while LIG4 can be partially stabilized by this mutant. The separation-of-function of this mutant could help identify the functions of Lig4 stabilizing activity of X4 and the roles of X4-XLF interaction. The authors further show that MEFs from X4M61R mice show mild sensitivity to IR and phleomycin, and X4M61R does not severely affect B- and T-lymphocyte development in mice. The authors therefore suggest that the X4-XLF interaction is not critical for V(D)J recombination in vivo. The authors further investigated the functional redundancies between PAXX, ATM, and X4 in lymphoid development by generating X4M61R-PAXX-/- and X4M61R-ATM-/- double mutant mice. The X4M61R-PAXX-/- and X4M61R-ATM-/- double mutant mice exhibit impaired V(D)J recombination, causing SCID phenotype. The authors further suggest that PAXX and ATM are compensatory factors of X4M61R in immune development. The authors also introduced X4M61R on XLF-/- background mice. They found that X4M61R-XLF-/- mice experience late embryonic lethality caused by massive neuronal apoptosis, and exhibit defect in V(D)J recombination.

1. In the Introduction section, the authors propose their idea of a "double DNA repair synapsis" model for repair of broken ends specifically caused by RAG1/2 complex during V(D)J recombination. However, the synapsis model is not clear here. The authors might want to provide more details about the model in this section, so readers could easily understand the model. The authors suggest that RAG2, PAXX, and ATM signaling mediate the first synapse to stabilize the RAG-cut broken ends; XRCC4-XLF filaments mediate the second synapse to tether the broken ends. The "two synapsis" model was proposed based on the observations of functional redundancy in V(D)J recombination between XLF and RAG2, PAXX, and ATM kinase activity. In the current manuscript, the authors seek to understand the contributions of XRCC4 to this model. However, the "two synapse" model is not very consistent with other studies. PAXX and XLF were reported to stabilize the synaptic complex of broken ends by interacting with Ku70/Ku80 complex (PMID: 29786079, PMID: 31399561, PMID: 33289484). This suggests that PAXX and XLF have overlapping functions in mediating end synapsis, and this does not mean PAXX and XLF are within different synaptic states (first synapse and second synapse). Moreover, if XRCC4 and XLF are in the same "second synapse" pathway, X4M61R-XLF-/- double mutant mice then should perform V(D)J recombination to some extent; this would lead to at least modest lymphocyte development, because the "first synapse" by either RAG2 or ATM, or PAXX could compensate for the "second synapse" as suggested by the authors. But X4M61R-XLF-/- mice reported by the authors are actually immune deficiency. The interaction of XLF and XRCC4 is indeed important for end synapsis, but it does not rely on the XRCC4-XLF filament. While the results here are interesting, the authors might need to rephrase the Introduction and frame the scientific question of this manuscript in a way that reconciles the above contradictions.

2. The authors conclude that PAXX/ATM are compensatory factors for X4 in immune development based on the SCID phenotype observed in X4M61R-PAXX-/- and X4M61R-ATM-/- double mutant mice. However, the results may also suggest that PAXX/ATM are compensatory factors for XLF but not directly for X4, because M61R mutation disrupts the interaction of X4 and XLF, which might result in XLF being unable to function in the X4M61R mice.

3. In lines 69-71, the authors cite Wang's paper to support the roles of XRCC4-XLF filaments in broken DNA end synapsis. However, in the Wang paper, XRCC4 and XLF are suggested to play roles mainly at the ends of the DNA (i.e., end-to-end synapsis) rather than through a XRCC4-XLF filament structure. Moreover, the statement of "X4 and PAXX binds DNA broken ends and stimulates X4-XLF DNA ends tethering in vitro" is not true. PAXX alone cannot bind dsDNA (as also reported in the cited Tadi et al., 2016 paper). It is true PAXX can stimulate end-to-end synapsis and ligation (PMID: 27705800, PMID: 31399561, PMID: 29786079). The stimulation by PAXX is through the interaction with Ku70/80, but not through the XRCC4-XLF filament.

4. The authors need to carefully examine their figures and related labels and legends to make sure they are correct. For example, one label is missing on Figure 1D; the x-axis labels of Figure 1G are incorrect; Figure 1 legend, line 134, "X4M61R vs X4-/- (bottom)" is inconsistent with those shown on Figure 1G.

5. The percentages of specific cell populations are shown on some flow cytometry panels but not on others. The ratios of the cell populations would make the flow cytometry data easier for readers to understand.

6. In line 164, the complete loss of viability was caused by 2 but not 1 Gy as shown on Figure 1G.

7. In line 228, the cellularity of X4M61R single mutant littermates in the spleen is missing.

8. The protein levels of X4 and Lig4 in the double knock-out mice/embryos could provide more information about the roles of X4 in lymphocyte development.

9. In the Introduction, the authors should also mention and consider the substantial literature supporting end-to-end synapsis models, which involve XLF interaction with XRCC4 and Ku. Both end-to-end synapsis and X4-XLF filament formation may be occurring.

10. The authors should consider that IR (e.g., 1 Gy) produces ~30 DSB per cell, whereas during V(D)J in T or B cells, there are likely to be only a few DSB per cell. One possible explanation for the difference between DNA repair roles and V(D)J is the number of events per cell and titration out of the XLF and X4 repair factors. Also, the chemistry and structure of the DNA ends for IR (and for phleomycin) are different than for V(D)J coding ends. The authors may want to consider that this could be an important reason for the difference in repair versus V(D)J for the role of X4.

*Reviewer #2 (Recommendations for the authors):*

XRCC4 (X4), a core component of the NHEJ DNA double strand break (DSB) repair pathway, functions to stabilize DNA Ligase IV (Lig4). X4 also associates with the XRCC4-like factor (Xlf) to form filaments that bridge and tether broken DNA ends. Both X4-/- and Lig4-/- mice show late embryonic lethality with massive neuronal apoptosis, as well as complete defects in V(D)J recombination. It was not known the extent that X4, aside from Lig4 stabilization, contributed to NHEJ in general DNA repair versus repair of programmed breaks during V(D)J recombination. To delineate differing functions of X4, a knock-in X4(M61R) mouse line was generated, where the M61R mutation was previously shown to disrupt interaction with Xlf while maintaining and stabilizing Lig4. A potential weakness of this model system is the reduced levels of X4(M61R) and Lig4 expression levels. Nevertheless, the overall results from this well-designed study are largely consistent with the authors' model and conclusions. Through combined genetic, immunological, and biochemical characterizations, the authors' illustrated three main points regarding X4 contributions to NHEJ. First, unlike X4-/-, X4(M61R) mice were viable with Lig4 stably expressed. Notably, while isolated X4(M61R) mouse cells showed deficiencies in repair of exogenously produced DNA DSBs, there were only relatively modest defects in B and T lymphocyte development, a marked contrast to that found with X4-/- mice. Second, The X4(M61R) mice phenocopy Xlf-/- mice by showing only minor defects in lymphocyte development, but with severe defects in V(D)J recombination when crossed with ATM or PAXX deficient mice. These results are consistent with the authors' model in which a "double DNA repair synapse" tethers the DNA ends in V(D)J recombination in a compensatory manner with one synapse mediated by the X4-Xlf filament and a separate synapse mediated by ATM signaling, PAXX, and the C-terminal region of RAG2. In a third point, X4(M61R) crossed with Xlf-/- mice showed late embryonic lethality due to post-mitotic neuronal apoptosis, with severe defects in V(D)J recombination in fetal hepatocytes and thymocytes. This latter point suggests that X4-Xlf filament cannot be compensated by a second DNA repair synapse in the fetal environment, perhaps due to the low expression of X4M61R. Alternatively, Xlf may function in an, as yet, unknown manner to rescue DNA repair defects of X4(M61R). In summary, using a novel separation of function X4 mutant, this study demonstrates that aside from its role in stabilizing Lig4, X4 is not requisite for B and T lymphocyte development likely due to functional redundancy of DNA end tethering in V(D)J recombination. Significantly, these results help to explain how patients with hypomorphic X4 mutations present with clear signs of DNA repair deficiency, but with no evident defects in development of the adaptive immune system.

This paper describes the development and characterization of a XRCC4 mutant mouse model, which helps to elucidate separate functions of X4 in NHEJ. Overall, this well-presented work demonstrates novel insights into the interplay between different NHEJ factors.

---

## [Author Response]

Essential revisions:1) Provide a clearer and more detailed introduction to models of functions of XRCC4, XLF etc in end joining that highlights the fundamental question(s) being addressed via generation of the mouse model described in this manuscript.

The Introduction was rephrased and reorganized as requested. We added in particular 3 references of recently published articles describing the cryo-EM structure of the NHEJ core complex, thus improving our knowledge on DNA synapsis during NHEJ mediated DNA repair. We hope it increased the understanding of the scope of our study and readability of the manuscript.

2) Both reviewers list several inconsistencies in the figures, such as labeling of axes, percentages of cell populations, splenic cellularity in certain mutant mouse lines, scales in Figure 4, mean/standard deviations etc. these should be carefully checked and made uniform throughout the manuscript.

We carefully checked the Figures and Figure legends for mistakes and inconsistency.

3) The protein levels of X4 and Lig4 in the double knock-out mice/embryos should be shown.

Unfortunately, we will not be able to address this issue in a reasonable time frame. Because of the successive lockdown due to covid19 pandemic, we have been asked to sacrifice most of our living animals. Since the study was essentially finished and the manuscript was being prepared we had no choice but kill the mice related to this project.

4) The Xlf-/- result is not shown in Figure 3E. Either the relevant citation needs to be included on this line, or the Xlf-/- result needs to be shown in Figure 3E.

Citation has been added.

The detailed comments from the reviewers are also appended below for providing context.Reviewer #1:XRCC4 deficient mouse model exhibits late embryonic lethality and defective lymphogenesis. However, no clinically significant immunodeficiency was observed in human patients with XRCC4 (X4) hypomorphic mutations, even in those with no detectable X4 in the fibroblasts. The contradictory phenomena observed in mouse and human make the roles of X4 in lymphocyte development unclear. Roch et al., developed a unique X4 mutant mouse model and investigated the roles of X4 in V(D)J recombination. The authors first generated a viable X4M61R knock-in mouse model using CRISPR/Cas9 technique. The M61R mutation disrupts the interaction between X4 and XLF, but not the interaction of X4 and LIG4. The expression of X4 protein is severely inhibited in the homozygous X4 M61R/M61R (X4M61R) mice, while LIG4 can be partially stabilized by this mutant. The separation-of-function of this mutant could help identify the functions of Lig4 stabilizing activity of X4 and the roles of X4-XLF interaction. The authors further show that MEFs from X4M61R mice show mild sensitivity to IR and phleomycin, and X4M61R does not severely affect B- and T-lymphocyte development in mice. The authors therefore suggest that the X4-XLF interaction is not critical for V(D)J recombination in vivo. The authors further investigated the functional redundancies between PAXX, ATM, and X4 in lymphoid development by generating X4M61R-PAXX-/- and X4M61R-ATM-/- double mutant mice. The X4M61R-PAXX-/- and X4M61R-ATM-/- double mutant mice exhibit impaired V(D)J recombination, causing SCID phenotype. The authors further suggest that PAXX and ATM are compensatory factors of X4M61R in immune development. The authors also introduced X4M61R on XLF-/- background mice. They found that X4M61R-XLF-/- mice experience late embryonic lethality caused by massive neuronal apoptosis, and exhibit defect in V(D)J recombination.1. In the Introduction section, the authors propose their idea of a "double DNA repair synapsis" model for repair of broken ends specifically caused by RAG1/2 complex during V(D)J recombination. However, the synapsis model is not clear here. The authors might want to provide more details about the model in this section, so readers could easily understand the model.

The Introduction was rephrased and reorganized to take into account these comment. We hope it increased the understanding of the scope of our study and readability of the manuscript.

The authors suggest that RAG2, PAXX, and ATM signaling mediate the first synapse to stabilize the RAG-cut broken ends; XRCC4-XLF filaments mediate the second synapse to tether the broken ends. The "two synapsis" model was proposed based on the observations of functional redundancy in V(D)J recombination between XLF and RAG2, PAXX, and ATM kinase activity. In the current manuscript, the authors seek to understand the contributions of XRCC4 to this model. However, the "two synapse" model is not very consistent with other studies.

As we mentioned in the introduction, the "two synapse" model is a strict specificity of the V(D)J recombination process (as opposed to random genotoxic-induced DSBs) because of the presence of the R1/2 post cleavage complex remaining on broken DNA ends as we proposed in our previous papers (Lescale et al., *Nat. Com*. 2016 and Betermier et al., *TCB* 2020). We believe it's the reason why it may appear sometimes inconsistent with purely DNA repair studies performed in vitro or *in cellulo* following genotoxics. The most striking inconsistency being the fact that although Xlf is a critical NHEJ factor as shown by the impressive radiosensitivity of Xlf deficient cells, the consequence on V(D)J recombination is almost null in contrast to what was well established for most NHEJ deficiencies.

PAXX and XLF were reported to stabilize the synaptic complex of broken ends by interacting with Ku70/Ku80 complex (PMID: 29786079, PMID: 31399561, PMID: 33289484). This suggests that PAXX and XLF have overlapping functions in mediating end synapsis, and this does not mean PAXX and XLF are within different synaptic states (first synapse and second synapse). Moreover, if XRCC4 and XLF are in the same "second synapse" pathway, X4M61R-XLF-/- double mutant mice then should perform V(D)J recombination to some extent; this would lead to at least modest lymphocyte development, because the "first synapse" by either RAG2 or ATM, or PAXX could compensate for the "second synapse" as suggested by the authors. But X4M61R-XLF-/- mice reported by the authors are actually immune deficiency. The interaction of XLF and XRCC4 is indeed important for end synapsis, but it does not rely on the XRCC4-XLF filament. While the results here are interesting, the authors might need to rephrase the Introduction and frame the scientific question of this manuscript in a way that reconciles the above contradictions.

The Introduction was rephrased and reorganized. We hope it increased the understanding of the scope of our study and readability of the manuscript.

2. The authors conclude that PAXX/ATM are compensatory factors for X4 in immune development based on the SCID phenotype observed in X4M61R-PAXX-/- and X4M61R-ATM-/- double mutant mice. However, the results may also suggest that PAXX/ATM are compensatory factors for XLF but not directly for X4, because M61R mutation disrupts the interaction of X4 and XLF, which might result in XLF being unable to function in the X4M61R mice.

We thank the Reviewer for this thought and added a sentence accordingly.

3. In lines 69-71, the authors cite Wang's paper to support the roles of XRCC4-XLF filaments in broken DNA end synapsis. However, in the Wang paper, XRCC4 and XLF are suggested to play roles mainly at the ends of the DNA (i.e., end-to-end synapsis) rather than through a XRCC4-XLF filament structure. Moreover, the statement of "X4 and PAXX binds DNA broken ends and stimulates X4-XLF DNA ends tethering in vitro" is not true. PAXX alone cannot bind dsDNA (as also reported in the cited Tadi et al., 2016 paper). It is true PAXX can stimulate end-to-end synapsis and ligation (PMID: 27705800, PMID: 31399561, PMID: 29786079). The stimulation by PAXX is through the interaction with Ku70/80, but not through the XRCC4-XLF filament.

The Introduction was rephrased and reorganized to take into account these comment. We hope it increased the understanding of the scope of our study and readability of the manuscript.

4. The authors need to carefully examine their figures and related labels and legends to make sure they are correct. For example, one label is missing on Figure 1D; the x-axis labels of Figure 1G are incorrect; Figure 1 legend, line 134, "X4M61R vs X4-/- (bottom)" is inconsistent with those shown on Figure 1G.

We carefully checked the Figures and Figure legends for mistakes and inconsistency.

5. The percentages of specific cell populations are shown on some flow cytometry panels but not on others. The ratios of the cell populations would make the flow cytometry data easier for readers to understand.

The % of cell population was shown only on Fig2D. For consistency with all other FACS panel we removed the % values on Fig2D. The actual values of gated cell population are computed on side histograms.

As suggested, we tried representation of the data by ratio rather than true values. As shown in Author response image 1 on two examples, we believe that the ratio representation (right) makes the data less readable in addition to losing the information of the real numbers (left) as it compresses the differences. We decided to keep data representation as it was*.*

**Author response image 1. sa2fig1:** 

6. In line 164, the complete loss of viability was caused by 2 but not 1 Gy as shown on Figure 1G.

Corrected.

7. In line 228, the cellularity of X4M61R single mutant littermates in the spleen is missing.

Mean+/-SEM were checked for consistency.

8. The protein levels of X4 and Lig4 in the double knock-out mice/embryos could provide more information about the roles of X4 in lymphocyte development.

Unfortunately, we will not be able to address this issue in a reasonable time frame. Because of the successive lockdown due to covid19 pandemic, we have been asked to sacrifice most of our living animals. Since the study was essentially finished and the manuscript was being prepared we had no choice but kill the mice related to this project.

9. In the Introduction, the authors should also mention and consider the substantial literature supporting end-to-end synapsis models, which involve XLF interaction with XRCC4 and Ku. Both end-to-end synapsis and X4-XLF filament formation may be occurring.

We added a paragraph on DNA end synapsis with reference to Zhao et al., for further detailed reading and the 3 references on resolution of NHEJ core complexes through cryo-EM.

10. The authors should consider that IR (e.g., 1 Gy) produces ~30 DSB per cell, whereas during V(D)J in T or B cells, there are likely to be only a few DSB per cell. One possible explanation for the difference between DNA repair roles and V(D)J is the number of events per cell and titration out of the XLF and X4 repair factors. Also, the chemistry and structure of the DNA ends for IR (and for phleomycin) are different than for V(D)J coding ends. The authors may want to consider that this could be an important reason for the difference in repair versus V(D)J for the role of X4.

We agree that there are major qualitative and quantitative differences beetween VDJ and IR generated DSBs. Nevertheless, again taking Xlf deficiency as an example the one V(D)J break is repaired in Xlf KO mice but not anymore in many Xlf/NHEJ (or RAG2^cc^) double KO conditions. So we do not think that the number of breaks or their nature is the central issue, but rather the existence of a V(D)J specific backup system for the repair of prDSB.